# Consumer Choices in the Bread Market: The Importance of Fiber in Consumer Decisions

**DOI:** 10.3390/nu13010132

**Published:** 2020-12-31

**Authors:** Marta Sajdakowska, Jerzy Gębski, Marzena Jeżewska-Zychowicz, Maria Królak

**Affiliations:** Department of Food Market and Consumer Research, Institute of Human Nutrition Sciences, Warsaw University of Life Sciences (WULS-SGGW), 159C Nowoursynowska Street, 02-776 Warsaw, Poland; jerzy_gebski@sggw.edu.pl (J.G.); marzena_jezewska_zychowicz@sggw.edu.pl (M.J.-Z.); maria_krolak@sggw.edu.pl (M.K.)

**Keywords:** consumer, bread, fiber

## Abstract

The aim of the current study was two-fold: (1) to identify consumer segments based on bread selection motives and (2) to examine differences between the identified segments in terms of perception of bread and bread with added fiber, and information on the food label. The data were collected using a CAPI (computer-assisted personal interview) survey on a sample of 1013 consumers. The k-means clustering method was used to identify four clusters of consumers, namely, Enthusiastic, Involved, Ultra-Involved, and Neutral. The Enthusiastic was the group that expressed the most positive opinions about the bread and about the addition of fiber to white bread. Moreover, they appreciated the most the information placed on the bread label. On the other hand, the Ultra-Involved and the Involved presented moderate opinions on these issues. In contrast, the consumers from the Neutral segment agreed the least with the opinion that white bread fortified with fiber is healthier and more expensive compared to white bread without added fiber. Consumers belonging to the Enthusiastic segment declared, to a greater extent than others, that cereal products with added fiber facilitate a healthy lifestyle and can reduce the adverse effects of an inadequate diet. The obtained results indicated that relatively positive opinions on the addition of fiber to white bread, including its benefits for health, are an opportunity to further develop the market of cereal products with added fiber. However, the information about bread on the label and its readability should meet the expectations of consumers who differ significantly in terms of their motives for choice. Both now and in the future, this aspect will be a challenge for food entrepreneurs and organizations that are engaged in the education and development of information aimed at consumers.

## 1. Introduction

One of the ingredients of food that may have a beneficial effect on human health is fiber. Dietary fiber is naturally present in most plant-derived foods such as whole-grain cereals, vegetables, fruits, and pulses. Increasingly, fiber-containing ingredients and extracted dietary fibers are seen not only as a means to boost fiber intake but also as a functional replacement for other ingredients, such as sugars [1].

It is recommended to eat a diet rich in fiber, because it is a desirable ingredient due to its beneficial effects on human health [2], and at the same time still in many consumer groups its average daily intake is insufficient compared to the recommended one [3,4,5]. This is due to the insufficient consumption of vegetables, fruits, and wholegrain cereal products [6]. In order to facilitate the increase in the amount of fiber in the diet, food products to which this ingredient is added are launched to the market, including products that naturally contain fiber. On the one hand such food is of great interest to consumers [7,8], and on the other hand it is not accepted by some of them [9,10]. The lack of acceptance is the result of the perception of high-fiber food as unsavory [11] and the opinion that this type of food is characterized by a higher price in comparison to similar low-fiber food [12] and insufficient knowledge on the positive effects of fiber on health [6,13,14]. Increasing the amount of fiber by adding it to products can enhance consumer acceptance of that product [11,15]. A growing awareness of a healthy lifestyle among consumers, and products containing ingredients that have a beneficial effect on health, e.g., bread containing fiber, can attract consumers’ attention [16]. However, some evidence from research is contradictory, and others do not fully explain the observed behaviors [3,4,5].

Consumers’ decisions in the food market are influenced by factors such as taste, smell, appearance, and texture [17,18], as well as functional features (e.g., packaging, convenience of use), nutritional value, availability, brand, price, own experience, and opinions of other people [19]. Factors perceived as having a positive effect on health also play an important role in decisions, e.g., the naturalness of the product or the applied production method to increase health values of food [20,21,22]. These factors also determine the acceptance of products with added fiber [16,23]. 

In order to better understand the determinants of the choice of products with the addition of fiber, factors should be taken into account that determine the choice of the base product and those that are more specific to the enriched product [23,24,25].

The available research shows that consumers consider differently the presence of fiber in food as well as the addition of fiber to food [23,26]. These differences may be determined by their involvement in information searching, including using the food label as a source of information [27,28]. Consumers are increasingly searching information on food products, including information on the label [12,29]. In 2011 the Regulation (EU) No 1169/2011 of the European Parliament and of the Council on the Provision of Food Information to Consumers was published [30]. It should be emphasized, however, that the information available to the consumer affects the perception of the product, but this influence is largely determined by the consumer’s interest in such information. The sociodemographic characteristics of consumers (age, gender, social status) determine the use and understanding of this information [19,31]. 

The interest in a product with changed composition also depends on the acceptance of the product as a carrier of the added ingredient [23]. Thus, the reasons for choosing bread may be important in determining the acceptance of a modified product, i.e., white bread with fiber added. Therefore, the aim of the current study was two-fold: (1) to identify consumer segments based on bread selection motives and (2) to examine differences between the identified segments in terms of perception of bread and bread with added fiber, and information on the food label.

## 2. Materials and Methods

### 2.1. Data Collection Process

The study sample (N = 1013) was collected through a cross-sectional quantitative survey under the Bioproduct project. This paper presents some of the findings from a larger study [32]. Selection criteria of the sample considered the representativeness of the Polish population due to the province and the quota character by gender, education, and place of residence. Those above the age of 21 years and individuals who met other recruitment criteria, that is, consumption of at least two slices of bread a day and full or partial responsibility for the family’s grocery shopping, qualified for the interview. The interviews were conducted face to face at respondents’ homes by a professional market research agency, respecting the ESOMAR (European Society for Opinion and Marketing Research) code of conduct using the CAPI (computer-assisted personal interview) technique.

### 2.2. Description of Questionnaire

The questionnaire used in the study was structured in a few main blocks and covered aspects such as:(1)Importance of bread selection factors (a question: How important are the following factors for you when buying bread, where 1 = not important at all and 5 = very important?): (1) freshness, (2) personal or family preference, (3) use-by date, (4) knowledge of the product, (5) taste, (6) price, (7) availability, (8) place of purchase, (9) seller’s opinion, (10) quality label, (11) information on the packaging, (12) bread packaging, (13) manufacturer/brand, (14) additives to bread (e.g., grains, bran), (15) calorific value, (16) product composition, (17) tenderness, (18) color, (19) smell, (20) crunchiness, and (21) general appearance;(2)Lifestyle self-assessment (a question: How do you assess yourself in terms of your lifestyle, where 1 = I totally disagree and 5 = I totally agree?): (1) family-oriented, (2) valuing tradition, (3) caring for their own health, (4) involved in professional work, (5) paying great attention to the naturalness of food, (6) physically active, and (7) with high ecological awareness;(3)Consumers’ opinion regarding bread (a question: To what extent do you agree with the following statements on bread, where 1 = I totally disagree and 5 = I totally agree?): (1) In order to improve the health-promoting benefits, fiber can be added to the bread, (2) I buy more expensive bread because I think that the price goes along with the quality, (3) the taste of bread is more important to me than its health-promoting benefits, (4) in my opinion, bread should be sold in packaging, and (5) it is important to consume enough bread;(4)The importance of food information, including the importance of information on the food label (a question: To what extent do you agree with the following statements, where 1 = I totally disagree and 5 = I totally agree?): (1) Information on product packaging is very important to me, (2) I compare information on product labels before I decide which product to choose, and (3) I compare labels to choose products with the highest nutritional value;(5)The importance of the information on the bread label (a question: How important is the following information on the bread label for you, where 1 = not important and 5 = very important?): (1) price, (2) shelf life, (3) product name, (4) product weight, (5) product composition, (6) producer, (7) information on health effects, (8) information on the fiber content, and (9) quality label;(6)Opinions on cereal products with added fiber (a question: To what extent do you agree with the following statements about grain products with added fiber, where 1 = I totally disagree and 5 = I totally agree?): (1) They facilitate a healthy lifestyle, (2) may reduce the adverse effects of an inadequate diet, (3) I can prevent disease by eating such products regularly, (4) there is a need to add fiber to cereal products, and (5) the addition of fiber to cereal products worsens their taste; and(7)Opinions on white bread with added fiber compared to the same bread without added fiber (a question: Please compare white bread with added fiber to the same bread without fiber, where 1 = I totally disagree and 5 = I totally agree): (1) is healthier, (2) is more expensive, (3) has a higher nutrient content, (4) is less calorific, (5) is more difficult to find in shops, (6) has better taste, and (7) looks worse.

### 2.3. Statistical Analysis

Analysis of the statements’ reliability in a question regarding the motives of bread choice was performed using the Cronbach coefficient alpha. The obtained value of the Cronbach coefficient alpha = 0.83 confirmed the right choice of questions for factor analysis. Consumer segmentation was preceded by a factor analysis with a varimax rotation of 21 statements. Based on eigenvalues (eigenvalues higher than 1), a 3-factor solution describing the phenomenon analyzed was suggested (Table 1). The factors obtained via PCA analysis explained 61.64% of total variation. Qualification for particular factors was based on the minimum value of factor loadings estimated at 0.5, with factor adequacy for the requirements of factor analysis as studied by the Kaiser–Mayer–Olkin measure (KMO). The KMO value indicating collective correlation of variables was 0.91, which clearly confirmed the logic behind using the variable reduction method. The identified factors presented in Table 1 were used for cluster analysis (segmentation). 

Factor scores were subsequently used for cluster analysis, with the aim of obtaining segments of consumers that were internally homogeneous and at the same time different from one another in terms of the choice of bread (Table 2). The division of consumers into segments was conducted in two stages. First, a cluster analysis using hierarchical methods was performed. In the second stage, the k-means clustering method was used to identify segments of consumers. In the k-means method (the k-means clustering algorithm is an unsupervised algorithm that is used to segment the interest area from the background), in order to increase its efficiency, the average values for individual clusters obtained using the hierarchical method were used as seeds.

Four well-separated clusters were obtained, which was confirmed by both statistics assessing the selection of clusters, such as CCC (cubic clustering criteria), pseudo F, pseudo T2, or ANOVA (analysis of variance) statistics comparing the average values of variables for individual clusters (Table 2). Sociodemographic variables such as gender, age, education, size of the place of residence, and subjective assessment of the financial situation were used to profile the clusters (Table 3). The independence χ^2^ test was used to assess the diversity of profile features between clusters. In all statements analyzed, statistically significant (*p* < 0.05) differences between mean scores, particularly clusters, were observed. Additionally, a post-hoc test (Waller–Duncan k-ratio *t*-test) was used to compare mean values of opinions between pairs of clusters.

The statistical analysis was carried out using the SAS 9.4 statistical package (SAS Institute, Cary, NC, USA).

Taking into account the motives for choosing bread, among the consumers in segment no. 1, the most important were the sensory and nutritional motives as well as the marketing motives. In the case of segment no. 2, the most important were the sensory and nutritional motives, similarly to segment no. 1, and the least important were the marketing motives. For consumers in segment no. 3, the most important were the practical motives. In the case of segment no. 4, the least important factors in the selection of bread were the practical motives as well as the sensory and nutritional motives, and only slightly less important compared to segment no. 1 were the marketing motives (Table 2).

As previously indicated, the non-hierarchical k-means clustering method led to the identification of four clusters: Enthusiastic (no. 1), Ultra-Involved (no. 1), Involved (no. 3), and Neutral (no. 4). 

## 3. Results

### 3.1. Description of the Sample and Clusters

The characteristics of the study sample are shown in Table 3. There were more women (53.4%) than men (46.6%). The youngest persons, i.e., persons under 30 years of age, constituted 1/5 of all respondents (20.9%), and the oldest persons, i.e., persons over 60 years of age, constituted more than 1/4 (26.3%). Respondents with at least secondary education (i.e., secondary, bachelor’s, engineering, and higher) accounted for about 65% of all the study sample. More than half of the surveyed persons came from cities. Respondents evaluating their material/financial situation at the moderate level (i.e., “We can afford some, but not all expenses”) constituted half of all respondents (53.5%).

In segment no. 1 (Enthusiastic; N = 241; 23.8%) there were more women and people who could meet all needs and even save than in other segments. Similar to segment no. 3, there were more people with higher education and, similar to segment no. 2, there were more people with secondary education and those aged 31–40. Among the Enthusiastic segment, there were the fewest people with vocational education and, similar to segment no. 3, the smallest number of people living in cities below 100,000 inhabitants.

In segment no. 2 (Ultra-Involved; N = 264; 26.0%) there were the least people aged 60 and more living in cities of 100,000–300,000 inhabitants as well as people with higher education, whereas the most people with secondary and vocational education lived in a city with less than 50,000 inhabitants. Similar to segment no. 4, there were few people with an income that would allow them to save.

In segment no. 3 (Involved; N = 193; 19.1%), similar to segment no. 4, there were more men. Moreover, compared to other segments, there were more people up to 30 years old, but also 51–60 years old and those living in the countryside, as well as more people with sufficient income to meet their needs. Similar to segment no. 1, there were more higher-educated people.

In segment no. 4 (Neutral; N = 315; 31.1%) there were the most men, similar to segment no. 3, and compared to other segments there were the most people aged 41–50 and over 60, and with elementary education. Moreover, there were a lot of people, similar to segment 2, with vocational education, mostly people from cities with 50,000–100,000 inhabitants, but also many from cities with 100,000–300,000 inhabitants, similar to segment 3. Among the Neutral group, there were the least people from cities over 300,000 inhabitants compared to other segments.

Table 4 shows the self-assessment of lifestyle in the study group. The respondents in general considered themselves to be the most family-oriented, tradition-oriented, and health-conscious; however, differences were observed between the segments. The Enthusiastic segment, to a greater extent than the other segments, perceived themselves mainly as people who valued tradition, took care of their own health, paid attention to the naturalness of food, were physically active, and had high ecological awareness. In the case of the Ultra-Involved segment, a fairly high level of compliance with most lifestyle statements was indicated. However, it should be emphasized that compared to the Enthusiastic and the Involved segments, they showed the lowest compliance with the statement indicating that they had high environmental awareness, similar to the Neutral segment. Involved respondents declared to the highest extent that they were family-oriented compared to consumers in other segments, as well as being engaged in professional work, similar to the Enthusiastic segment. Neutral respondents perceived themselves as the least family-oriented and traditional, as well as people who took care of their own health and engaged in professional work. They also showed the same level of compliance with the statements indicating that they paid great attention to the naturalness of food and were physically active, similar to the Ultra-Involved and the Involved segments.

### 3.2. General Opinions on Bread and Information on the Food Label from a Consumer Point of View

In Table 5 respondents’ opinions on bread are presented. Respondents most agreed with the statement indicating that fiber can be added to bread in order to improve health-promoting benefits. Moreover, respondents showed a high level of compliance with the statement indicating that information on the product’s packaging is very important to them.

The Enthusiastic respondents rated most of the statements higher than the other segments. Nevertheless, they were similar to the Ultra-Involved and the Involved respondents in assessing the statement indicating that the taste of bread is more important for them than its health-promoting benefits. Furthermore, similar to the Neutral respondents, they stated that it was important to consume enough bread.

In the case of the Involved segment, the lowest level of compliance with the opinion was recorded regarding the purchase of more expensive bread due to the idea that the price goes along with the quality, and toward the statement that it is important to eat enough bread as well as with regard to the comparison of labels in order to choose products with the highest nutritional value. 

Similar to the Ultra-Involved, the Involved indicated the lowest score for the statement “In my opinion, bread should be sold in packaging”. Moreover, similar to the Ultra-Involved and the Neutral, the Involved reported the same level of compliance with the statement indicating that “Information on product packaging is very important to me.”

The respondents of the Neutral and Ultra-Involved segments showed the lowest level of compliance in relation to the opinion on adding fiber to bread in order to improve its health-promoting benefits. The respondents from the Ultra-Involved and Involved segments indicated the lowest ratings for compliance with the necessity to sell bread in packaging.

The opinions on the information placed on bread labels are shown in Table 6. The three most important pieces of information were price, shelf life, and the name of the product. 

The Enthusiastic segment showed the highest ratings for most of the information on the bread label compared to the other segments, with the exception of the product name. The Ultra-Involved declared lower ratings for most of the information placed on bread labels compared to the Enthusiastic, but significantly higher than the other two segments. The statement “Information on health effects” was rated lower by segment nos. 2, 3, and 4 compared to 1. For the Neutral segment, price, shelf-life information, and product name were the least important, whereas for the Involved, the composition of the product, information on the fiber content, and the quality label were the least important. Moreover, there were no differences between segment nos. 3 and 4 regarding the rating for the statements “Product weights” and “Producer.”

### 3.3. The Importance of Adding Fiber to Cereal Products

Respondents’ opinions on the importance of adding fiber to cereal products are shown in Table 7. The respondents most strongly agreed with the statements that such products facilitate a healthy lifestyle and that they can reduce the adverse effects of an inadequate diet. The Enthusiastic segment declared to a greater extent than others that cereal products with added fiber facilitate a healthy lifestyle, can reduce the adverse effects of an inadequate diet, as well as that diseases can be prevented by eating such products regularly and that there is a need to add fiber to cereal products. On the other hand, people in the Neutral segment agreed to the lowest extent with the opinion that cereal products with added fiber may reduce the adverse effects of an inadequate diet. The Neutral segment, similar to the Enthusiastic segment, believed that the addition of fiber to cereal products worsens their taste. The Involved consumers expressed the lowest level of compliance with the statement informing about worsening taste of cereal products after adding fiber in comparison with the other segments. The Involved and the Neutral as well as the Ultra-Involved showed no differences in opinion about the need to add fiber to cereal products.

The opinions concerning the comparison between white bread with added fiber to white bread without added fiber are shown in Table 8. The people surveyed most agreed with the statement that white bread with added fiber is healthier and has a higher nutrient content, but also claimed that it is more expensive compared to bread without the addition of fiber. According to the Enthusiasts, white bread with added fiber is healthier, has a higher content of nutrients, is less calorific, and has a better taste compared to white bread without added fiber. However, in their opinion, this type of bread is also more difficult to find in stores. Similar to the segments of the Ultra-Involved and the Neutral, the Enthusiastic segment indicated that this kind of bread looks worse compared to the same bread without fiber.

The group of the Involved segment, to the lowest degree in comparison with the other segments, agreed with the opinion that bread with added fiber has a better taste and looks worse than bread without the addition of fiber. On the other hand, the Ultra-Involved, to the lowest degree, agreed with the opinion that it has higher nutritional content in comparison with bread without the addition of fiber. Consumers identified as Neutral least agreed with the opinion that white bread with added fiber is healthier, as well as being more expensive and less calorific than bread without fiber.

## 4. Discussion

The differences in the factors determining the choice of staple foods [20,33] suggest that they cannot be ignored in research on the acceptance of reformulated foods. Thus, the research was aimed at identifying groups of consumers differing in terms of their motives when choosing bread. As a result of the analysis of 21 motives, four segments were identified. Respondents qualified for particular segments showed differences in the motives for choosing bread, but also in the sociodemographic characteristics.

The obtained results confirmed the existence of differences in the motives for choosing bread. It was shown that the marketing motives were less important among the Ultra-Involved and the Involved, i.e., in segments that were quite diversified in terms of sociodemographic characteristics. The similarity between the Enthusiastic and the Ultra-Involved segments concerned the importance of sensory and nutritional aspects. Their common feature was their age (31–40 years old), and they were also people with different education and income, coming from cities of different sizes.

Thus, the sociodemographic specificity of the identified segments is visible, which confirms the benefits of using segmentation in explaining differences in food acceptance and consumer behavior [34,35]. Thanks to such knowledge, it should be easier and more effective to reach consumers with marketing communication. Despite the fact that the motives taken into account refer to food available on the market, supplementing the results of this segmentation with other motives or characteristics may significantly increase the knowledge about consumer behavior towards new products, e.g., with a modified composition.

It is expected that health-related motives will play an increasingly important role when a food choice is made, because among the values appreciated by consumers, health is perceived as the most important value [36,37,38]. Positive attitudes towards health among some consumers are associated with a pro-healthy dietary pattern characterized by the consumption of plant-origin foods (fruit and vegetables) [39]. Moreover, consumers search for food with specific health values [40,41]. Opinions concerning the health aspects resulting from the presence of fiber in the product are confirmed in the literature [7,16]. Our research also shows that the surveyed people most agreed with the statements that cereal products with added fiber facilitate a healthy lifestyle and that they can reduce the adverse effects of an inadequate diet. The Enthusiastic segment perceived these benefits the most, and the Neutral the least. Furthermore, the consumers of the Enthusiastic segment were the greatest supporters of adding fiber to bread to improve its health-promoting benefits, with the Neutral and the Ultra-Involved disagreeing the most with it.

Although the consumption of bread is decreasing, consumers still perceive it as a staple food [42,43,44]. Bread can play a significant role among products with positive health benefits for the consumer, e.g., wholemeal bread [45], and all the more so since some consumers accept increasing the level of fiber in bread [8,46]. Increasing the amount of fiber in the white bread can be especially important for people who cannot or do not want to eat wholegrain bread. Nevertheless, adding fiber to white bread influences not only its appearance and other sensory attributes, but also its price, which may be a barrier to purchasing it [4,47]. Our research shows that in order to improve the health benefits, fiber can be added to bread. This opinion was presented especially by the Enthusiastic segment. Similarly, other studies confirmed that some consumers appreciate the value of fiber [5], however, there is still a group of consumers with a low level of knowledge of its positive role in the diet [6,13,14]. The differences in accepting the addition of fiber are reflected in the identified segments. The Enthusiastic segment, i.e., mainly women aged 31–40, better educated (secondary and higher education), from larger cities, and with the higher income, most appreciated the benefits of adding fiber to bread, although, similar to the Neutral segment, those consumers most noticed the deterioration of taste resulting from the addition of fiber. The Neutral were the most skeptical about cereal products with added fiber and they were most similar to the Ultra-Involved. Results of previous studies also indicated that for some cereal products, i.e., muffins prepared from white flour fortified with fiber and wholemeal flour, consumers were not always convinced that the fiber content could be high enough to encourage them to buy a product fortified with fiber. As a result, consumers rated such products with added fiber lower [11].

The sensory, nutritional, and marketing motives that were important to the Enthusiastic segment are therefore associated with the greater acceptance of adding fiber to cereal products. In order to reduce consumers’ doubts about products with added fiber, it is necessary to inform them about the product in an understandable and acceptable way. Generally, information on cereal products [6,48,49], including the information on the label, is important for consumers [20,50]. However, studies also indicated that consumers do not always read the information on the label, e.g., due to lack of time or too much information. Moreover, it was observed that food labelling is more useful for specific consumer groups, such as athletes, consumers with health conditions, or consumers concerned with a healthy lifestyle [28]. Our study confirmed the differences in the perception of information on the label. To interact effectively with the consumer, information on the label, its legibility, and the effect of fiber should be understandable enough for consumers to highlight the benefits and encourage them to choose a product with increased fiber content [11]. However, it is important to point out that EU food law does not include fiber in its list of mandatory back-of-pack nutrients and does not allow fiber to be included in the front-of-pack declaration—hindering opportunities to signal the fiber content of products [51,52]. In addition, the literature also identifies a number of barriers that limit consumers’ fiber intake (e.g., relating to health claims or cooking skills), which may, however, result in consumers seeking products with increased fiber levels [53,54].

At the same time, it should be emphasized that the Enthusiastic segment surveyed was the group that expressed the most positive opinions about the bread, the importance of the information placed on the bread label, as well as about the addition of fiber to white bread. Among the Enthusiastic segment, women prevailed, and, according to literature studies, women have more favorable attitudes towards health [55], as well as the role of fiber in the diet [20]. Comparing white bread with added fiber to white bread without the addition of fiber, the people surveyed most agreed with the claim that it is healthier and has a higher content of nutrients, but also claimed that white bread with added fiber is more expensive compared to bread without the addition of fiber. As before, the Enthusiastic segment agreed with most of the statements proposed in the study. In contrast, consumers in the Neutral segment, agreed the least with the opinion that white bread fortified with fiber is healthier and more expensive compared to white bread without added fiber. Previous research indicated that some consumers did not perceive significant differences between the white and the wholemeal bread products; which may be due to a lack of nutritional awareness associated with the fiber and other ingredients found in whole grain flour. This group of consumers was not motivated to change their eating behaviors [56]. On the other hand, some studies showed that consumers recognize products that are rich in fiber, and those with whole grains [7,57].

Studies by Cattaneo et al. (2019) showed that consumers appreciated receiving information that may facilitate their purchasing decision related to foods produced with some technologies, since this seemed to increase their confidence in foods [58]. Literature studies also indicated that food shoppers of the future will be time optimizers; more health-conscious, with both health and well-being growing in importance; more individual and open to a more personalized experience but only if there is a clear benefit to them with minimum effort; more experimental; and more socially conscious with regard to sustainability and ethical choices, although price, availability and quality will still come first [52].

The strength of our results is the relatively large sample from the Polish population. Nevertheless, the findings have some limitations. The sample included only those solely or jointly responsible for the family’s grocery shopping. The cross-sectional design of the study and data collection at a single point in time did not allow for conclusions to be drawn about causality, but rather only about the associations between variables. 

The study considered survey data that included some bias. The first concerns self-reported information obtained from the questionnaire that may be inaccurate, due to the unnatural situation created by the questionnaire itself. Moreover, the real environment in which the choice of bread takes place and the use of the products themselves rather than using the survey/questionnaire would provide a better reflection of the conditions under which purchasing decisions are made. However, research using the real product or labels was not logistically or economically possible in this part of study, considering the size of the sample. Due to the abovementioned limitations, our findings should be used with caution with populations with different cultural backgrounds. Despite these limitations, our study provides new insights into determinants of bread choice.

## 5. Conclusions

The obtained results indicate that relatively positive opinions on the importance of the addition of fiber in the diet and the acceptance of its addition to white bread, including its benefits for health, are an opportunity to further develop the market of cereal products with added fiber. However, the differences in motives of bread choice, as well as in opinion on fiber addition to bread, were observed.

The Enthusiastic was the group that expressed the most positive opinions about bread, the importance of the information placed on the bread label, as well as about the addition of fiber to white bread. In contrast, consumers in the Neutral segment agreed the least with the opinion that white bread fortified with fiber is healthier, more expensive, and less calorific compared to white bread without added fiber.

Moreover, information about bread on the label and its readability should meet the expectations of consumers. Both now and in the future, this aspect will be a challenge for food entrepreneurs and organizations that are engaged in the education and development of information aimed at consumers. Furthermore, paying attention to the marketing aspects that were important for the Enthusiastic as well as for the Neutral groups indicated that producers and sellers should particularly take into account communication with the consumers, including information on labels when developing policies and strategies, in order to increase consumer interest regarding the marketing of cereal products, e.g., bread with added fiber.

Therefore, these results are of relevance for professionals involved in public nutrition issues as well as for marketers aiming at the development of well-tailored communication strategies. The results may also provide important insights for those who develop educational strategies and campaigns.

## Figures and Tables

**Table 1 nutrients-13-00132-t001:** Principal components analysis (PCA) of consumers’ use of bread selection motives; varimax rotated factor loadings percentage of explained variance (N = 1013, Poland).

Bread Selection Motives	The Familiarity and Availability of Product—The Practical Motives Factor 1	The Importance of Producer, Seller, and Information—The Marketing Motives Factor 2	The Sensory and Composition Features of Product—The Sensory and Nutritional Motives Factor 3
Freshness	0.771		
Personal or family preference	0.768		
Use-by date	0.750		
Knowledge of the product	0.692		
Taste	0.685		
Price	0.584		
Availability	0.560		
Place of purchase		0.766	
Seller’s opinion		0.706	
Quality label		0.679	
Information on the packaging		0.652	
Bread packaging		0.619	
Manufacturer/brand		0.590	
Additives in the bread (e.g., grains, bran)		0.513	
Calorific value			0.664
Product composition			0.660
Tenderness			0.642
Colour			0.636
Smell			0.599
Crunchiness			0.587
General appearance			0.505
The variance explained/% explained variance	37.70	13.33	10.61

**Table 2 nutrients-13-00132-t002:** Characteristics of the identified segments according to the PCA factors (motives of bread selection); the mean ratings of the segments on the classification variables.

Factors/Bread Selection Motives	Segment 1	Segment 2	Segment 3	Segment 4	*p*-Value
The familiarity and availability of product—the practical motives	3.76 ^b^	2.36 ^c^	4.27 ^a^	1.92 ^d^	<0.0001
The importance of producer, seller, and information—the marketing motives	4.38 ^a^	1.79 ^c^	1.73 ^c^	3.78 ^b^	<0.0001
The sensory and composition features—the sensory and nutritional motives	4.15 ^a^	4.13 ^a^	2.02 ^b^	1.83 ^c^	<0.0001

Different superscripts indicate significantly different means following the ANOVA post hoc Waller–Duncan test.

**Table 3 nutrients-13-00132-t003:** Socio-demographic characteristics of the consumers surveyed (N = 1013, Poland).

Variables	Total Sample	Segment 1	Segment 2	Segment 3	Segment 4	*p*-Value
Sex						0.0004
Female	53.4	63.9	55.3	47.7	47.3	
Male	46.6	36.1	44.7	52.3	52.7	
Age						0.0353
Up to 30 years	20.9	19.9	21.3	23.8	19.6	
31–40 years	17.9	20.7	23.5	15.0	12.7	
41–50 years	16.0	14.5	13.6	14.0	20.3	
51–60 years	18.9	19.5	18.9	21.3	17.2	
Over 60 years	26.3	25.4	22.7	25.9	30.2	
Education						0.0002
Elementary	6.1	5.8	5.7	3.6	8.2	
Vocational	29.4	20.7	34.5	25.4	34.3	
Secondary	36.5	39.5	38.6	35.7	33.0	
Bachelor’s/Engineer	9.5	8.7	7.6	10.9	10.9	
Higher	18.5	25.3	13.6	24.4	13.6	
Place of residence						<0.0001
Village	38.4	40.3	37.1	41.5	36.2	
Town below 50,000	16.3	9.9	26.9	12.4	14.6	
Town from 50,000 to 100,000	13.9	10.8	13.6	11.9	17.8	
City from 101,000 to 300,000	18.7	19.9	8.7	23.3	23.2	
City from 301,000 to 500,000	5.8	6.6	6.9	5.7	4.4	
City over 500,000	6.9	12.5	6.8	5.2	3.8	
Opinion on family income						0.0283
Is not sufficient at all	5.7	4.5	4.2	4.1	8.9	
Allows to meet only basic needs	26.7	24.9	26.9	22.3	30.8	
We can afford some, but not all expenses	53.5	52.3	56.8	57.0	49.5	
We can afford everything	11.2	13.3	10.2	13.5	8.9	
We can afford everything, and in addition we can save	2.9	5.0	1.9	3.1	1.9	

χ^2^ test of independence, *p*-value < 0.05—differences between groups are significant.

**Table 4 nutrients-13-00132-t004:** Profile of segments in terms of statements referring to self-evaluation of one’s own lifestyle.

Statements	Mean	Enthusiastic 1	Ultra-Involved 2	Involved 3	Neutral 4	*p*-Value
Family-oriented	4.04	4.01 ^b^	3.98 ^b^	4.34 ^a^	3.65 ^c^	<0.0001
Valuing tradition	3.95	4.32 ^a^	3.95 ^b^	4.03 ^b^	3.64 ^c^	<0.0001
Caring for their own health	3.81	4.11 ^a^	3.81 ^b^	3.85 ^b^	3.58 ^c^	<0.0001
Involved in professional work	3.57	3.91 ^a^	3.50 ^b^	3.89 ^a^	3.17 ^c^	<0.0001
Paying great attention to the naturalness of food	3.56	4.05 ^a^	3.48 ^b^	3.61 ^b^	3.24 ^b^	<0.0001
Physically active	3.56	3.86 ^a^	3.55 ^b^	3.51 ^b^	3.40 ^b^	<0.0001
With high ecological awareness	3.36	3.79 ^a^	3.14 ^c^	3.38 ^b^	3.22 ^bc^	<0.0001

Different superscripts indicate significantly different means following the ANOVA post hoc Waller-Duncan test.

**Table 5 nutrients-13-00132-t005:** Profile of segments in terms of statements referring to bread and to information on food.

Statements	Mean	Enthusiastic 1	Ultra-Involved 2	Involved 3	Neutral 4	*p*-Value
In order to improve the health-promoting benefits, fiber can be added to the bread	3.50	3.84 ^a^	3.33 ^c^	3.58 ^b^	3.34 ^c^	<0.0001
I buy more expensive bread, because I think that the price goes along with the quality	3.38	4.00 ^a^	3.22 ^b^	3.04 ^c^	3.29 ^b^	<0.0001
The taste of bread is more important to me than its health-promoting benefits	3.27	3.31 ^ab^	3.30 ^ab^	4.09 ^a^	3.16 ^b^	0.0483
In my opinion, the bread should be sold in packaging	3.19	3.54 ^a^	2.95 ^c^	2.83 ^c^	3.36 ^b^	<0.0001
It is important to consume enough bread	3.12	3.14 ^ab^	3.09 ^b^	2.87 ^c^	3.30 ^a^	<0.0001
Information on product packaging is very important to me	3.51	3.97 ^a^	3.36 ^b^	3.37 ^b^	3.40 ^b^	<0.0001
I compare information on product labels before I decide which product to choose	3.35	3.78 ^a^	3.30 ^b^	3.30 ^b^	2.29 ^c^	<0.0001
I compare labels to choose products with the highest nutritional value	3.27	3.67 ^a^	3.25 ^b^	2.77 ^c^	3.31 ^b^	<0.0001

Different superscripts indicate significantly different means following the ANOVA post hoc Waller-Duncan test.

**Table 6 nutrients-13-00132-t006:** Profile of segments in terms of statements referring to information on the bread label.

Statements	Mean	Enthusiastic 1	Ultra-Involved 2	Involved 3	Neutral 4	*p*-Value
Price	4.33	4.78 ^a^	4.30 ^c^	4.60 ^b^	3.86 ^d^	<0.0001
Shelf life	4.30	4.81 ^a^	4.20 ^c^	4.60 ^b^	3.81 ^d^	<0.0001
Product name	4.05	3.98 ^b^	3.94 ^b^	4.64 ^a^	3.76 ^c^	<0.0001
Product weight	3.95	4.57 ^a^	3.90 ^b^	3.66 ^c^	3.70 ^c^	<0.0001
Product composition	3.89	4.67 ^a^	3.87 ^b^	3.34 ^d^	3.67 ^c^	<0.0001
Producer	3.81	4.55 ^a^	3.66 ^b^	3.50 ^c^	3.56 ^bc^	<0.0001
Information on health effects	3.79	4.63 ^a^	3.53 ^b^	3.45 ^b^	3.58 ^b^	<0.0001
Information on the fiber content	3.64	4.51 ^a^	3.53 ^b^	2.84 ^c^	3.57 ^b^	<0.0001
Quality label	3.61	4.54 ^a^	3.44 ^b^	2.79 ^c^	3.57 ^b^	<0.0001

Different superscripts indicate significantly different means following ANOVA post hoc Waller-Duncan test.

**Table 7 nutrients-13-00132-t007:** Profile of segments in terms of statements referring to cereal products with added fiber.

Statements	Mean	Enthusiastic 1	Ultra-Involved 2	Involved 3	Neutral 4	*p*-Value
They facilitate a healthy lifestyle	3.69	4.07 ^a^	3.58 ^bc^	3.70 ^b^	3.50 ^c^	<0.0001
May reduce the adverse effects of an inadequate diet	3.60	4.05 ^a^	3.57 ^c^	3.75 ^b^	3.21 ^d^	<0.0001
I can prevent disease by eating such products regularly	3.52	3.95 ^a^	3.41 ^bc^	3.50 ^b^	3.30 ^c^	<0.0001
There is a need to add fiber to cereal products	3.51	3.90 ^a^	3.43 ^b^	3.43 ^b^	3.34 ^b^	<0.0001
The addition of fiber to cereal products worsens their taste	2.95	3.02 ^ab^	2.88 ^b^	2.68 ^c^	3.13 ^a^	<0.0001

Different superscripts indicate significantly different means following ANOVA post hoc Waller-Duncan test.

**Table 8 nutrients-13-00132-t008:** Profile of segments in terms of statements referring to white bread with added fiber compared to the same bread, but without fiber.

Statements	Mean	Enthusiastic 1	Ultra-Involved 2	Involved 3	Neutral 4	*p*-Value
Is healthier	3.71	4.17 ^a^	3.76 ^b^	3.85 ^b^	3.24 ^c^	<0.0001
Is more expensive	3.62	3.92 ^a^	3.53 ^b^	3.91 ^a^	3.31 ^c^	<0.0001
Has a higher nutrient content	3.61	4.07 ^a^	3.51 ^c^	3.61 ^b^	3.74 ^b^	<0.0001
Is less calorific	3.48	3.93 ^a^	3.42 ^b^	3.51 ^b^	3.20 ^c^	<0.0001
Is more difficult to find in shops	3.37	3.59 ^a^	3.32 ^b^	3.28 ^b^	3.30 ^b^	0.0025
Has better taste	3.36	3.92 ^a^	3.29 ^b^	2.92 ^c^	3.26 ^b^	<0.0001
Looks worse	2.91	3.00 ^a^	2.95 ^a^	2.47 ^b^	3.08 ^a^	<0.0001

Different superscripts indicate significantly different means following ANOVA post hoc Waller-Duncan test.

## Data Availability

Not applicable.

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
