# Peer review of "Consumer Choices in the Bread Market: The Importance of Fiber in Consumer Decisions"

_nutrients, 2020, doi:10.3390/nu13010132_

Round 1
Reviewer 1 Report
In my opinion, the entire article moves on some basic equivocations that leave the reader rather puzzled as to the meaning that the article itself intends to have. It is probably the research questions themselves that cause these misunderstandings. What does "to identify the importance of fibre in the diet" mean? Is it a health food article? to which "..and the importance of the food label" is added: but, how can you keep these two aspects in the same research question? Furthermore, it seems absolutely pointless to make an article to confirm the two statements: it is out of the question that 1) fibre is important for a healthy and balanced diet, and 2) food labeling is important. Is there a single author who denies it?
But the article doesn't focus on that. So let's go to the first research question: "to identify consumer segments based on bread selection motives and to examine differences between the identified segments in terms of perception of white bread and white bread fortified with fibre". Here, too, we note, first of all, that the questions proposed are two and not just one. But the thing that generates the most confusion, and that makes the writings of the authors sometimes bordering on banality, is this: if in substance, the survey is used to identify consumer segments based on their choices, what is the meaning of a discussion that merely reverses the same relationship? In terms of extreme simplicity, if I identify the Enthusiastic consumer segment based on their preference for white bread with added fibre, what is the point of reiterating that Enthusiastic pay more attention to the fibre diet? Because in essence this is exactly what the article does most of all. The first thing to do is therefore to clarify the research questions and the objectives of the article. Furthermore, it should be better specified how the four (why not three?) categories of consumers have been identified, the statistical reference is not enough.
The development of the article, and in particular the comments on the various tables, must also be reviewed. Again, the feeling is of comments that vary between being either trivial or completely obvious. Some naivety must also be excluded: for example, at the beginning of the Introduction, the factors that influence consumers' decisions on the food market are listed: well, apart from the fact that there is no mention of advertising and proximity to the point of purchase, but is it ever possible not to include the price?
The four subdivision segments of the sample (Enthusiastic, Ultra-Involved, Involved, Neutral) arouse many doubts, above all because in many cases the expected coherence of the answers to the various statements schematized in the tables is completely lost. In truth, there seems to be more similarity between the first segment and the third (and sometimes even the fourth), rather than between the first and the second: for example, in tables 3, 6 and 7 it is so for almost all of the statements contained. The surprising thing is that these apparent inconsistencies are not commented on and / or explained by the authors. A similar thing happens, for example, in the comment lines of Table 2 (in many ways the most important of all). Here the authors dwell on affirmations that add nothing on the peculiarities of the four segments with respect to the total sample (for example it is affirmed that "In segment 3 ... almost half of the respondents were consumers aged 50 and over", which in fact does not differentiate at all the segment with respect to the total sample), while nothing is said about the real 'surprises' that emerge from the data shown (for example, with reference to the level of education, the 'strange' similarity between segments one and three, and two and four).
In short, and in conclusion, it seems to me that the large amount of data collected (1013 interviews conducted face-to-face at the respondents' homes really seem a huge amount of perception data to be used) is (for now) a lost opportunity to give really interesting answers to appropriate research questions.
Author Response
Reviewer 1
In my opinion, the entire article moves on some basic equivocations that leave the reader rather puzzled as to the meaning that the article itself intends to have. It is probably the research questions themselves that cause these misunderstandings. What does "to identify the importance of fibre in the diet" mean? Is it a health food article? to which "..and the importance of the food label" is added: but, how can you keep these two aspects in the same research question?
We agree that the way of wording the research questions/aims could lead to the misunderstanding of the paper concept. The comment was taken into account and the objectives were reformulated.
The final aim of the paper is: to identify consumer segments based on bread selection motives and (2) to examine differences between the identified segments in terms of perception of bread and bread with added fibre, and information on food label.
Therefore, we concentrated on distinguishing consumer segments according to the motives for choosing bread, and on checking whether the motives that may be specific to the choice of bread with added fibre (perception of the importance of fibre in the diet, searching for information about fibre on the product label, but also the specificity of lifestyle and socio-demographic features) differentiate the distinguished segments.
If so, it can be valuable information for interested people, i.e. food producers (not only the product development but also marketing messages), educators, etc.
Furthermore, it seems absolutely pointless to make an article to confirm the two statements: it is out of the question that 1) fibre is important for a healthy and balanced diet, and 2) food labeling is important. Is there a single author who denies it?
We agree with the comment that researchers do not deny the importance of fibre in a balanced diet or information on the label. We also do not deny to these aspects in this particular area.
However, people perceive these issues differently. Depending on the perception of fibre, the consequences of adding it to the product, or the information on the label, these people may be more or less interested in purchasing a product with a modified composition. Therefore, the variables relating to these issues were included in the study.
But the article doesn't focus on that. So let's go to the first research question: "to identify consumer segments based on bread selection motives and to examine differences between the identified segments in terms of perception of white bread and white bread fortified with fibre". Here, too, we note, first of all, that the questions proposed are two and not just one.
It has been corrected. As was mentioned above (the first part of the reply),the first aim was to identify consumer segments based on bread selection motives, and the second was to examine differences between the identified segments by taking into account the variables describing the perception of a fibre addition, searching for information about fibre on the product label, but also lifestyle aspects and socio-demographic characteristics.
But the thing that generates the most confusion, and that makes the writings of the authors sometimes bordering on banality, is this: if in substance, the survey is used to identify consumer segments based on their choices, what is the meaning of a discussion that merely reverses the same relationship? In terms of extreme simplicity, if I identify the Enthusiastic consumer segment based on their preference for white bread with added fibre, what is the point of reiterating that Enthusiastic pay more attention to the fibre diet? Because in essence this is exactly what the article does most of all.
We appreciate this suggestion. In the current version of the article, there is a table (table no. 2) showing the results of segmentation, the basis of which were the factors identified in the PCA (Principal Component Analysis; 21 themes taken into account when selecting bread).
The differences between the mean values of the factor in individual segments were confirmed using ANOVA; as the next step the characteristics of individual segments was made, taking into account the variables concerning the perception of bread with added fibre, information about fibre on the label, etc. (Tables 4-8). Therefore, we hope that this change and this clarification makes the discussion more legitimate and can be more understandable.
The first thing to do is therefore to clarify the research questions and the objectives of the article.
As suggested, the aims of the paper have been clarified.
Furthermore, it should be better specified how the four (why not three?) categories of consumers have been identified, the statistical reference is not enough.
In the current version of the article, there is a table showing the results of segmentation, the basis of which were the factors distinguished in the PCA (21 themes taken into account when selecting the bread), then normalized by the range method (from the mean value subtracted and divided by SD). The normalized values of the factors were divided into 5 quintiles (1-5); the means obtained are from quintiles. The differences between the segments were confirmed by the use of the formal ANOVA test with the post hoc test.
The development of the article, and in particular the comments on the various tables, must also be reviewed. Again, the feeling is of comments that vary between being either trivial or completely obvious. Some naivety must also be excluded: for example, at the beginning of the Introduction, the factors that influence consumers' decisions on the food market are listed: well, apart from the fact that there is no mention of advertising and proximity to the point of purchase, but is it ever possible not to include the price?
It was corrected. According to this suggestion the comments on the tables were reviewed. We hope we managed to avoid the obvious comments. Moreover, as was mentioned previously there was included a change in the introduction in order to create justification for the research and the method of analyzing the obtained empirical material.
The four subdivision segments of the sample (Enthusiastic, Ultra-Involved, Involved, Neutral) arouse many doubts, above all because in many cases the expected coherence of the answers to the various statements schematized in the tables is completely lost. In truth, there seems to be more similarity between the first segment and the third (and sometimes even the fourth), rather than between the first and the second: for example, in tables 3, 6 and 7 it is so for almost all of the statements contained.
As mentioned before, the segments were distinguished based on the bread selection factors and the differences between the segments were confirmed (Table no. 2). The similarity of the segments disclosed in the description of the results showed that the respondents differed in terms of the motives for choosing the bread, but showed similar characteristics after taking into account the characteristics describing the segments. Changes have been made to the paper so that the reader does not get confused while reading.
The surprising thing is that these apparent inconsistencies are not commented on and / or explained by the authors. A similar thing happens, for example, in the comment lines of Table 2 (in many ways the most important of all). Here the authors dwell on affirmations that add nothing on the peculiarities of the four segments with respect to the total sample (for example it is affirmed that "In segment 3 ... almost half of the respondents were consumers aged 50 and over", which in fact does not differentiate at all the segment with respect to the total sample), while nothing is said about the real 'surprises' that emerge from the data shown (for example, with reference to the level of education, the 'strange' similarity between segments one and three, and two and four).
The description of the results has been revised taking into account the comments in the review. The segments were described in order to show not only the differences but also the similarities between the segments. Therefore, the final description will be more understandable.
In short, and in conclusion, it seems to me that the large amount of data collected (1013 interviews conducted face-to-face at the respondents' homes really seem a huge amount of perception data to be used) is (for now) a lost opportunity to give really interesting answers to appropriate research questions.
We hope that the corrections that were included in the final version of the paper made this reviewer's opinion not current actually.
We would like to kindly thank the Reviewer for the time and effort taken to read and review our article. We appreciate the constructive criticism and all suggestions.

Reviewer 2 Report
Thank you and the opportunity to read the material.
The introduction is too short. There is an incomprehensible introduction to the topic. Are we talking about consumer behavior or about fiber? There is no link that would connect these two trends.
Material and methods: Please attach the questionnaire in the supplementary documents. The analytical part of the paper will be more understandable.
Please correct the tables - they go beyond the page boundaries.
What were the limitations of the study performed?
The Strengths and Limitations section is missing.
Please complete this section with specific practical tips for the entities mentioned in the conclusion: food entrepreneurs and educational organizations.
The footnotes should be adapted to the requirements of the publisher.
Author Response
Reviewer 2
The introduction is too short. There is an incomprehensible introduction to the topic. Are we talking about consumer behavior or about fiber? There is no link that would connect these two trends.
There was made a change/correction in the introduction in order to create justification for the research and the method of analyzing the obtained empirical material. The subject of the research are the determinants of consumer behaviour in the food market, but special attention is paid to fibre as a food ingredient, especially in the form added to food. We hope that the verified version of the Introduction justifies the research undertaken and does not raise any doubts related to the essence of the research problem.
Material and methods: Please attach the questionnaire in the supplementary documents. The analytical part of the paper will be more understandable.
In order to make the presented results more understandable, the questions were included in the methodology (Description of Questionnaire). We hope that this form of presentation the research questions will meet the reviewer's expectations. In our opinion, this part of the paper is more clear now.
Please correct the tables - they go beyond the page boundaries.
In line with this suggestion, the tables have been corrected so that they do not exceed the margin range.
What were the limitations of the study performed? The Strengths and Limitations section is missing.
The Strengths and Limitations were introduced to the manuscript.
Please complete this section with specific practical tips for the entities mentioned in the conclusion: food entrepreneurs and educational organizations.
Specific practical tips for the entities mentioned in the conclusion: food entrepreneurs and educational organizations were introduced to the manuscript.
The footnotes should be adapted to the requirements of the publisher.
The footnotes were changed.
Thank to Reviewer for all indicated suggestions.
We would like also to thank for the time and effort taken to read and review our article.

Reviewer 3 Report
Review of Consumer Choices in the Bread Market: The Importance of Fibre in Consumer Decisions
This paper uses a random sample of consumers in Poland to examine the perceived importance of fiber in bread. The paper uses principle components and cluster analysis methods to identify four segments of consumers based on their attitudes toward bread fiber. The paper finds that fiber is important to a large segment of the population and identifies several sociodemographic variables that correlate to consumers’ perceptions of fiber.
I found the paper interesting. I am not an expert on principal-components analysis, but the quantitative work appears to be adequately done.
I do suggest the authors consider changing their aim (2) on line 77. The aim as currently written is, “(2) to identify the importance of fibre in the diet…….” I don’t think this is the authors aim. Instead, I think the authors intend to identify how consumers perceive fiber as important in their diets.
Starting in table 3 and continuing through table 7, the authors present mean ratings for different consumer segments. I found the authors’ method of designating statistically significant differences between means to be confusing. The table annotates means with letters, a, b, or c. Apparently, if segments have mean ratings with the same letter then the difference between those means is statistically equal to zero. Does that mean that if letters are different, then means are not equal? Probably, but I’m not sure. It is not discussed in the paper. And, how are we to determine whether the mean for a segment is statistically different from the overall mean. For example, in Table 3, the mean rating for “Family-oriented” is 4.04. The mean for Enthusiastic is 4.01 annotated with a b. The mean for Ultra-involved is 3.98, again annotated with a b. So the mean ratings for Enthusiastic and Ultra-involved are equal, but are they statistically different than the overall mean of 4.04? We don’t know.
Similarly, in the same table what does the P-value at the end of the row indicate? I suggest clarifying the information in tables 3-7; I found them confusing.
I also suggest explaining the relative sizes of the different segments. If I read correctly, 23.8% of consumers are “Enthusiastic” and 26% are “Ultra-Involved”. What is the reader to make of these percentages? If you have a representative sample, does this mean that 49.8% of the population perceives fiber in bread to be important? Or, is it the case that the segmenting method naturally tends to find approximately equal sized segments? Without an explanation, it is easy for reader to misinterpret the significance of the sizes of the segment. I’m guessing the size of the segments does not tells us that much, but I could be wrong.
Author Response
Reviewer 3
This paper uses a random sample of consumers in Poland to examine the perceived importance of fiber in bread. The paper uses principle components and cluster analysis methods to identify four segments of consumers based on their attitudes toward bread fiber. The paper finds that fiber is important to a large segment of the population and identifies several sociodemographic variables that correlate to consumers’ perceptions of fiber.
I found the paper interesting. I am not an expert on principal-components analysis, but the quantitative work appears to be adequately done.
We appreciate this comment very much.
I do suggest the authors consider changing their aim (2) on line 77. The aim as currently written is, “(2) to identify the importance of fibre in the diet…….” I don’t think this is the authors aim. Instead, I think the authors intend to identify how consumers perceive fiber as important in their diets.
Thank you for this suggestion. The change of the research goal was introduced to the text, also taking into account the comments of another reviewer.
Starting in table 3 and continuing through table 7, the authors present mean ratings for different consumer segments. I found the authors’ method of designating statistically significant differences between means to be confusing. The table annotates means with letters, a, b, or c. Apparently, if segments have mean ratings with the same letter then the difference between those means is statistically equal to zero. Does that mean that if letters are different, then means are not equal? Probably, but I’m not sure. It is not discussed in the paper. And, how are we to determine whether the mean for a segment is statistically different from the overall mean. For example, in Table 3, the mean rating for “Family-oriented” is 4.04. The mean for Enthusiastic is 4.01 annotated with a b. The mean for Ultra-involved is 3.98, again annotated with a b. So the mean ratings for Enthusiastic and Ultra-involved are equal, but are they statistically different than the overall mean of 4.04? We don’t know.
Usually, in the segmentation the mean scores in the segments are compared using analysis of variance (ANOVA for 3 or more segments).
The most popular approach is to compare the mean scores between the segments, so the means for the analyzed features/items in individual segments are not compared to the means for the total sample.
Similarly, in the same table what does the P-value at the end of the row indicate? I suggest clarifying the information in tables 3-7; I found them confusing.
P-value shows the significance level; in general we adopted a significance level of <0.05.
For ANOVA, the hypotheses are as follows:
H0: all analyzed means did not differ statistically
H1: at least 1 mean differs statistically from the others, of course with the adopted significance level of 0.05.
In the case of the indicated tables, the p-value is in all cases less than 0.05, so we reject the H0 hypothesis in favor of its H1 alternative (at least 1 mean differs from the others).
Then the post hoc test (Waller-Duncan test) compared the individual means among themselves also at the significance level of 0.05. Various indices prove that there is a statistically significant difference between these means, and the same letter indicates no statistically significant difference between these means.
I also suggest explaining the relative sizes of the different segments. If I read correctly, 23.8% of consumers are “Enthusiastic” and 26% are “Ultra-Involved”. What is the reader to make of these percentages? If you have a representative sample, does this mean that 49.8% of the population perceives fiber in bread to be important? Or, is it the case that the segmenting method naturally tends to find approximately equal sized segments? Without an explanation, it is easy for reader to misinterpret the significance of the sizes of the segment. I’m guessing the size of the segments does not tells us that much, but I could be wrong.
We appreciate this suggestion.
Actually, the method of extracting clusters (Euclidean distances) is not intended to aim at a similar number of clusters.
In fact, the size of the cluster does not tell too much to the reader, however, usually this piece of information is used in order to show the size of each segment (usually by number or by percentage). So, we also decided to indicate the percentage of the clusters/segments to make the paper more clear for the final reader.
Therefore, we hope that this form of presentation will meet the reviewer's expectations.
Thank to Reviewer for all indicated suggestions.
We would like also to thank for the time and effort taken to read and review our article.

Round 2
Reviewer 1 Report
Most of my comments have been considered by the authors, who have made major changes to their work. The article now appears to me significantly improved. I suggest the authors a careful final re-reading aimed at simplifying the writing, where possible, which is now sometimes burdened by the insertions made.
I still have some perplexities: I still believe that better thought out research questions could have better exploited the available data, just as some 'curiosities' emerged from the data would have deserved more attention. However, on the whole, I now consider the work done interesting and acceptable.
Reviewer 2 Report
Dear authors,
In the text, reference numbers should be placed in square brackets [ ], and placed before the punctuation; for example [1–3] or [1,3].
I wish you next success.